# Efficacy of the Chinese version interpretation bias modification training in an unselected sample: A randomized trial

Fan Zhang[1�} , Chenwei Huang[2�} , Xiaofei Mao[1�} , Tianya Hou[1], Luna Sun[1], Yaoguang Zhou[1], Guanghui Deng[1]*

1 Faculty of Psychology, Naval Medical University, Shanghai, China, 2 School of Basic Medical Science, Naval Medical University, Shanghai, China

} These authors contributed equally to this work.
* bfbedu@126.com

**Data Availability Statement:** All relevant data are within the paper and its Supporting Information files.

## Abstract

Training individuals to interpret ambiguous information in positive ways might be an effective method of reducing social anxiety. However, little research had been carried out in Chinese samples, and the effect of interpretation training on other processes such as attentional bias also remained unclear. This study examined the effect of interpretation bias modification program (IMP) on interpretation bias, social anxiety and attentional bias, and the possible mediation effects. 51 healthy adults were randomly assigned to either a 5-session IMP training that guided them to endorse benign interpretation in ambiguous scenarios or an interpretation control condition (ICC). Self-reported measures of social anxiety symptoms, attentional bias and interpretation bias were evaluated before and after training. Results showed that compared to control group, IMP group generated more positive interpretations and less negative interpretations after training ($F(1,49) = 7.65$, $p<0.01$, $\eta_p^2 = 0.14$; $F(1,49) = 14.60$, $p<0.01$, $\eta_p^2 = 0.23$ respectively). IMP yielded greater interpretation bias reduction ($F(1,49) = 12.84$, $p<0.01$, $\eta_p^2 = 0.21$) and social anxiety reduction ($F(1,49) = 21.39$, $p<0.01$, $\eta_p^2 = 0.30$) than ICC, but change in attentional bias was not significant between IMP and the control group. Change in interpretation bias did not show a significant mediation effect in the relationship between training condition and social anxiety reduction. This study provided preliminary evidence for the efficacy of the Chinese version of IMP training. Possible methodological issues and interpretations underlying the findings were discussed. This study was registered in Chinese Clinical Trial Registry (www.chitr.org.cn), a WHO approved registry. The title of registration trial was "A Study on the efficiency of cognitive bias and attentional bias training on fear and phobia" and the registration number was ChiCTR2100045670.

## Introduction

Theoretical models have suggested that biased threat-related information processing is associated with the occurrence and development of social anxiety [1]. Two typical types of the biased processing are attentional bias (AB), a bias to selectively attend to threat cues in the

**Funding:** This work was funded by the Major Program of the "13th Five-Year Plan" for Medical Development of PLA (BWS16J012) and Scientific research project of Shanghai Health Commission (20204Y0287). The funding agencies had no role in the analysis and interpretation of data; in the writing of the manuscript; or in the decision to submit the paper for publication.

**Competing interests:** The authors have declared that no competing interests exist.

environment in socially anxious individuals [2], and interpretation bias (IB), a bias to interpret negative meanings in ambiguous social situations in social anxiety [3]. Fortunately, AB and IB are malleable with Cognitive Bias Modification (CBM) training. In Attentional Bias Modification (ABM) and CBM for interpretation biases (CBM-I), participants are taught to preferentially attend to, process, or engage in specific types of stimuli (i.e., positive, neutral), and simultaneously avoid others (i.e., negative, threatening). Meta analyses of Randomized Controlled Trials (RCT) have provided early evidence for efficacy of ABM and CBM-I in reducing cognitive biases and social anxiety symptoms, but the effects were moderated by characteristics of the training procedure and study design [4, 5].

There are currently three main types of CBM-I: the homograph paradigm [6], the word-sentence association task (WSAT) paradigm [7], and the ambiguous situations paradigm [8]. WSAT paradigm-based CBM-I, also called Interpretation Modification Program (IMP), is a computerized intervention that modifies interpretation biases by providing positive feedbacks when participants make benign interpretations in ambiguous scenarios and providing negative feedbacks in response to threat interpretations of ambiguous scenarios. RCTs of IMP have suggested its efficacy in reducing social anxiety symptoms. For example, Beard and Amir showed that IMP effectively reduced IB and social anxiety symptoms in a subclinical sample [7]. Later, Amir and Taylor [9] reported significant decreases in social anxiety symptoms after IMP training in a clinical sample with generalized social anxiety disorder. In their study, 65% of the participants in the training group no longer met clinical diagnostic criteria after training compared with 13% in the control group. Meta-analyses have showed that CBM-I could reduce interpretation biases and anxiety [5, 10] and CBM-I might be more effective in reducing anxiety symptoms than ABM [11]. Although the merits of CBM-I are well-documented, there is currently few Chinese version of CBM-I available which limits the application of this emerging psychological intervention method in China.

CBM was hypothesized to improve symptoms by changing the biases that were targeted [12], and researchers have investigated possible mechanisms of modification of interpretation bias on social anxiety. For example, Mobini et al. [13] have showed that the change in positive interpretations could predict the change in social anxiety. In their study, the change in positive interpretations was influenced by both the decrease in negative interpretations and the increase in positive interpretations. Naim et al. [14] found that the change in interpretation bias did not mediate the indirect effect of treatment on the reduction of social anxiety symptoms. In this study, the change in interpretation bias was also a reflection of both changes in positive interpretations and negative interpretations. In other studies, the changes in negative and positive interpretations were separately calculated to reveal distinct effects. Bread and Amir [7] have found the indirect effect of training conditions on social anxiety symptoms through an increase in benign interpretations. Amir and Tylor [9] have shown that the reductions in endorsement of threatening interpretations were associated with a greater decrease in social anxiety symptoms, but the increase in endorsement of benign interpretations was not linked to the decrease in social anxiety. These results indicated that negative and benign interpretations might have different effects on social anxiety reduction, but given the current mixed results, this conclusion needed to be further confirmed.

From a theoretical perspective, the different types of information processing biases in anxiety might share a common mechanism [15]. In the first published research, White and colleagues [16] reported that the change in attentional bias (AB) affected the manner in which ambiguous information was interpreted, showing the "cascading effect". This result was also confirmed in other studies [17]. However, it is still unclear whether the manipulation of interpretation bias could affect the attentional bias. Amir, Bomyea and Beard [18] have showed that AB was changed after a single session of IMP. But in a later study, multi-session CBM-I

training did not result in the reduction of AB, and moreover, the combination of attentional and interpretation bias modification was less effective than attentional bias modification alone [14]. Modification of interpretation bias has been shown to improve attentional control [19], which played an important role in AB in social anxiety [20]. Thus, it was reasonable to predict a change in AB after CBM-I training and further research was needed.

To conclude, the present study aimed to evaluate the efficacy of a Chinese version IMP designed to change interpretation bias and social anxiety. As a primary study, healthy adults with a willingness to change social anxiety symptoms were recruited and randomly assigned to the intervention or control condition. To explore the possible mechanism of IMP, self-generated positive and negative interpretations in ambiguous scenarios were calculated to estimate the relationship between interpretation changes and social anxiety symptoms. AB was also tested before and after IMP training. The hypotheses were (i) Chinese version IMP was an effective method to reduce negative interpretation bias and social anxiety in healthy adults, (ii) IMP improved social anxiety by reducing negative interpretations or/and by increasing positive interpretations and (iii) the manipulation of interpretation bias could change the attentional bias.

## Methods

### Trial design and any changes after trial commencement

In this study, participants were randomly assigned between IMP intervention and ICC training with a 1:1 allocation ratio (see Fig 1).

### Participants, eligibility criteria and settings

This study was conducted at the faculty of psychology, Naval Medical University in October, 2019. A total of 51 eligible subjects (28 males and 23 females) with the average age of 21.78 ±2.52 participated in the present study. Individuals participated in the experiment should meet the following inclusion criteria: 1) subjective complaints about social anxiety problems; 2) 18 years of age or older; 3) comfortable using computer.

The study was approved by the ethics review committee of Naval Medical University in accordance with the ethical standards established in the 1964 Declaration of Helsinki and its later amendments. Participants signed informed consents prior to their participation in the study.

### Intervention

The design of IMP was similar with the previous study [18]. As shown in Fig 2, a fixation was presented for 500ms and was replaced by a threat word or a benign word presenting for 1000ms. Then a description of an ambiguous scenario was presented. Participants were asked to judge whether the scenario and the word were related or not. In the intervention condition (IMP group) the participants would receive a positive feedback if they decided that a benign interpretation and the scenario were related or that a threat interpretation and the scenario were unrelated. In the interpretation control condition (ICC group), feedback contingency was changed to 50% toward threat and 50% toward benign interpretations. Participants saw a total of 110 ambiguous scenarios and each paired with a threat and a benign word. Thus 220 word-sentences pairs were presented in each session. A single session lasted for 25–30 mins and participants were asked to complete one session a day for 5 days. Before and after the 5-day training, outcome variables were measured (see Fig 3). The training materials were the same with the previous study [7] and the materials were translated into Chinese. Several sentences were revised in accordance with Chinese culture.

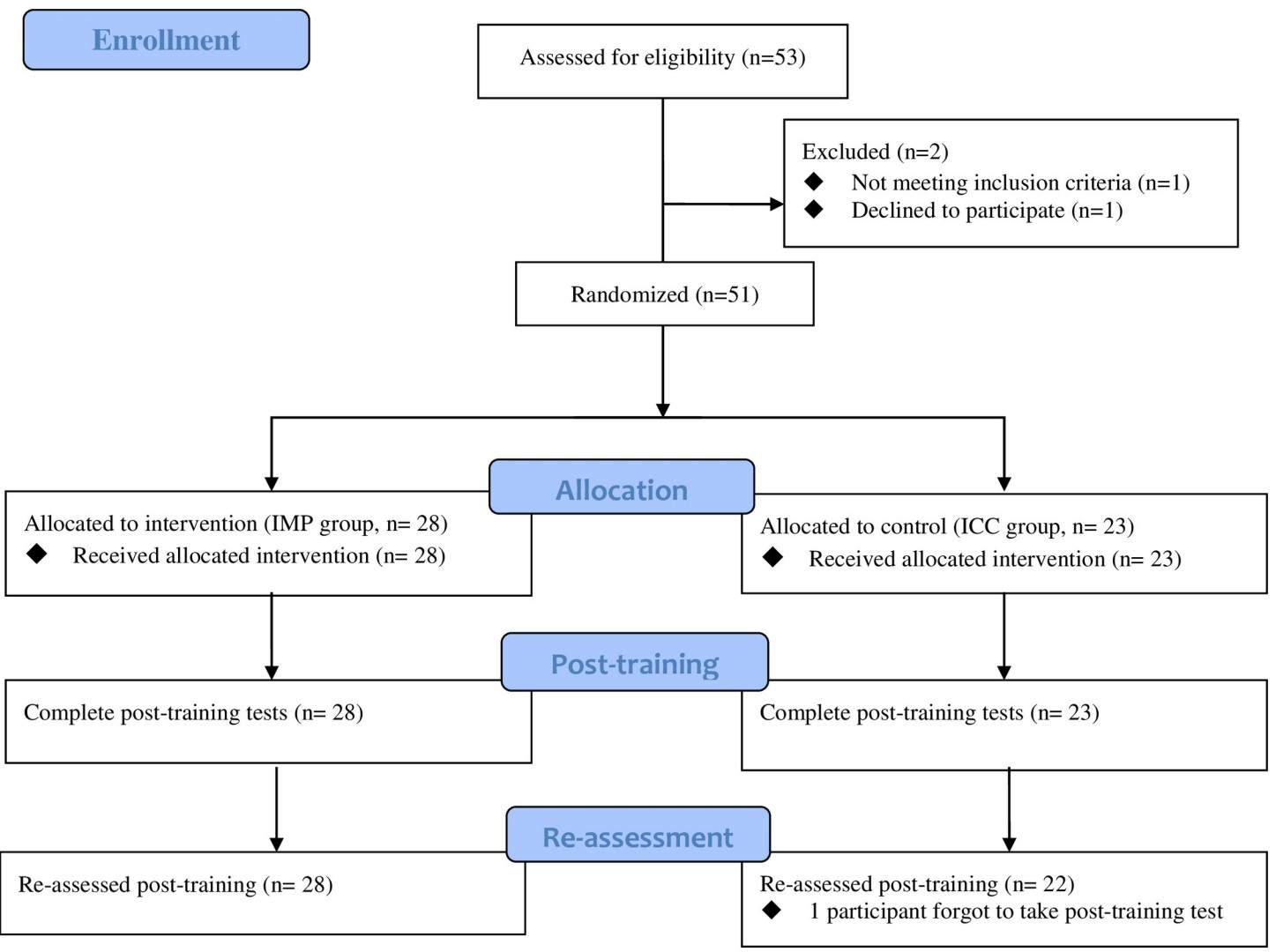

**Fig 1. Flowchart depicting the passage of participants.**

## Outcomes

**Self-report measures.** Interaction Anxiousness Scale served as the primary outcome measure. The IAS was a 5- point (1 = not at all, and 5 = extremely) Likert-type self-report measure assessing social anxiety symptoms. There were 15 items in the scale and the total score ranged from 15 to 75. The IAS has demonstrated good reliability and validity in Chinese undergraduates [21]. In the present study, Cronbach's Alpha at baseline was 0.83. Participants also completed the Depression Anxiety Stress Scale [22], which was widely used to test the effect of training [23]. The DASS was a 4-point (0 = not at all, and 3 = extremely) Likert-type self-report measure with a total of 21 items. In the present study, only the anxiety subscale (DASS_A, 7 items) was used, and Cronbach's Alpha of DASS_A at baseline was 0.65.

**Attentional bias assessment.** To measure threat-related attentional bias, we used a word-based dot-probe task adapted from the Tel-Aviv University/ National Institute of Mental Health Attention Bias Measurement Toolbox (TAU-NIMH ABMT) Initiative protocol (http://people. socsci.tau.ac.il/mu/anxietytrauma/research/). We replaced the threat and neutral words with

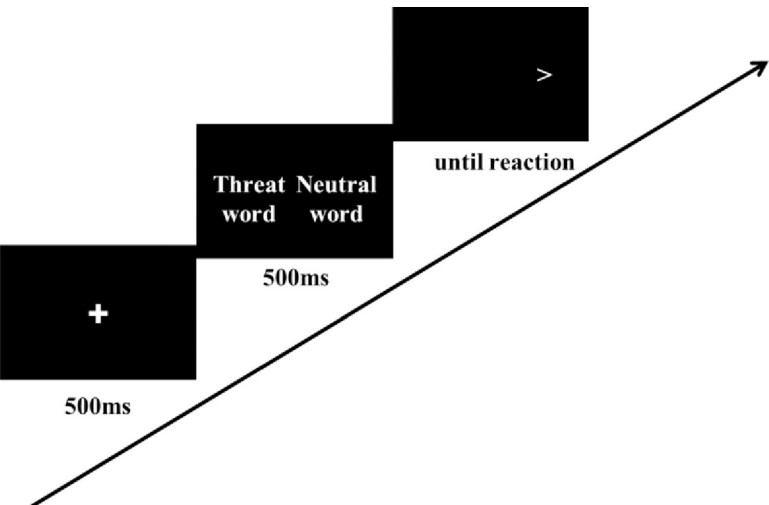

**Fig 2. Sequence of events in the dot-probe task.**

Chinese words from Chinese Affective Words System [24]. The valence of negative words was 2.78±0.04 and valence of neutral words was 5.58±0.07, showing a significant difference ($t(39) = 227.27, p<0.01$). The dot-probe task consisted of 120 trials: 80 threat-neutral and 40 neutral-neutral, and each word appeared twice. As shown in Fig 4, each trial began with a fixation presented for 500ms, then a pair of threat-neutral/ neutral-neutral words was shown for 500ms. Following the removal of the words, a probe ("<"or ">") appeared in the location of either the neutral or threatening word. The participants were instructed to determine the orientation of the probe by clicking the left or right mouse button. The probe remained on screen until the response was detected and then the next trial began. Threatening word location, probe location, and probe type were fully counterbalanced in presentation. In line with Cai et al. [23], only the reaction from correct hits was used in the following analyses and RTs <300ms or >1200ms were excluded. Mean threat-incongruent RT minus mean threat-congruent RT provided a measure of threat-related attention bias, such that positive values indicated the bias toward threat.

**Interpretation bias assessment.** The sentence completion task (SCT) adapted from Naim et al.'s study [14] was used to assess participants' interpretation bias. SCT was chosen because

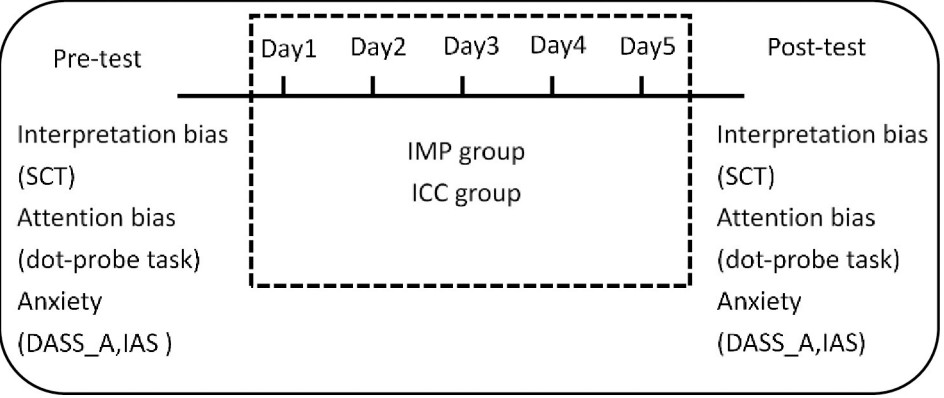

**Fig 3. Repeated measure study procedure.** SCT = sentence completion task; IAS = interpersonal anxiety scale; DASS_A = anxiety subscale in depression anxiety stress scale; IMP = interpretation modification program; ICC = interpretation control condition.

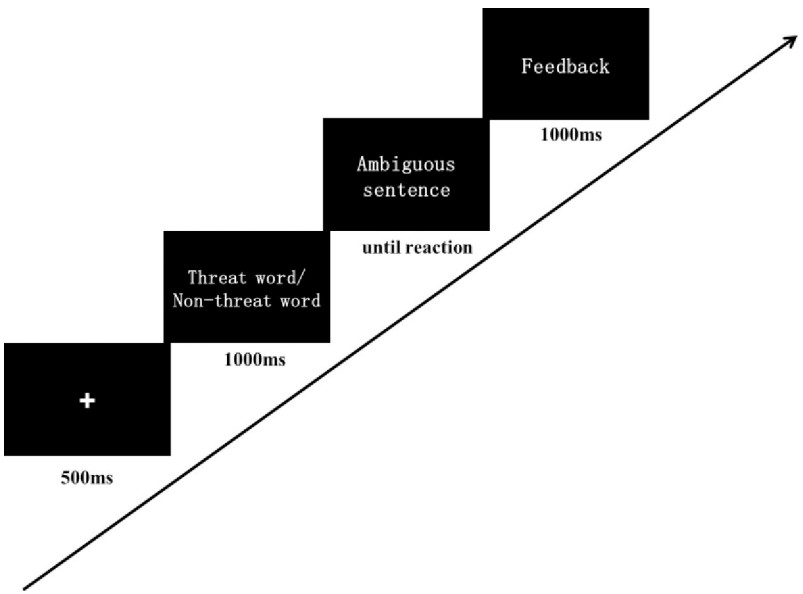

**Fig 4. Sequence of event in the IMP training.**

it was a holistic and ecologically valid approach [25]. The SCT consisted of 18 sentences describing ambiguous scenarios with the lasting word missing (e.g. "The tutor called you to the office, you feel ____."). Adding the final word could determine explicitly the valence of the sentence. Subjects were asked to imagine themselves in the situation and generate as many endings as possible for each sentence. Then, participants coded each of their sentence completion as positive, negative or neutral but they did not know how the scores would be calculated. Three interpretation scores were calculated: SCT_P and SCT_N represented the average number of positive and negative endings of each sentence; SCT represented the average value of [(positive endings—negative endings)/sum of all endings of each sentence]. Thus, negative interpretation bias was represented by the negative values of SCT.

## Sample size calculation

G*Power software version 3.1.9.2 was used to estimate required sample size for this study. This study used repeated measurement analysis of variance (ANOVA) to analyze time, group and interaction effects of the interventions. Therefore F-test (ANOVA repeated measurement, within-between interactions) was chosen. Sample size was based on detecting changes in social anxiety levels. In line with the previous study [9], effect size ($\eta_p^2$) was set at 0.23 and alpha value was set at 0.05. Approximately 40 participants (i.e., 20 participants in each arm) would provide 80% power and 52 participants (i.e., 26 participants in each arm) would provide 90% power to detect a statistical significance.

## Interim analyses and stopping guidelines

Not applicable.

## Randomization and blinding

Randomization in 1:1 allocation was determined by the following process: upon arriving at the laboratory, participants were informed of their numbers which were generated automatically

by the computer. The numbers (ranged from 1 to 100) were generated by the random number function of the Microsoft Excel software. The participants and the trial staffs (SL and ZY) know the numbers, but they were blind to the allocation. In fact, if the number was an odd number, the participant was assigned to the ICC group and if the number was an even number, the participant was assigned to the IMP group. Allocation concealment was kept until interventions and post-training tests were finished.

## Data analysis

Data were analyzed using SPSS version 19. As shown in Fig 1, one participant forgot to take the post-training tests. To address missing values and being consistent with intention-to-treat principles [26], replacement values of post-training data of this participant were generated using multivariate imputation by chained equations based on five replications. This method was applied because it led to less biased/more accurate results relative to single imputation, last observation carried forward, and complete case analysis [27]. Accuracy of two participants in the dot-probe task were under 50% and their data were deleted in the analysis of attentional bias.

Shapiro-Wilk normality tests were used to test data distribution and results showed that pre- and post-training DASS_A scores showed negative skewed distribution ($Z = 0.16$, $p<0.01$ for pre-training DASS_A; $Z = 0.14$, $p<0.05$ for post-training DASS_A). As a result, Box–Cox transformations [28] were performed where the $\lambda = -1$. The other variables were normally distributed and repeated measures of variance analyses ANOVAs were used to explore the time (pre-training, post-training) × condition (IMP, ICC) interaction effect on (i) interpretation bias, (ii) attentional bias, and (iii) social anxiety. Pre-training and post-training variables were compared between the two groups using independent *t*-test (two-tailed). Paired t-tests were conducted to compare post-assessment data to pre-assessment data. We calculated the change of interpretation bias, attentional bias and social anxiety by subtracting pre-treatment scores from post-treatment scores. Bootstrapping mediation analyses [29] were used to test possible effects of treatment on social anxiety symptom reduction through the change in interpretation bias.

## Result

### Interpretation bias

Table 1 presents the means and standard errors of interpretation bias, attentional bias and anxiety. To test the training effect of IMP on interpretation bias, the mean values of SCT were

**Table 1. Means and standard deviations for cognitive bias and social anxiety for intervention and control group.**

|  | Male%(N = 51) | Age(N = 51) | SCT (N = 51) | SCT_P (N = 51) | SCT_N (N = 51) | AB (N = 49) | IAS (N = 51) | DASS_A (N = 51) |
|---|---|---|---|---|---|---|---|---|
| Pre-training |  |  |  |  |  |  |  |  |
| ICC | 56.52% | 21.74±2.65 | 0.09±0.56 | 6.43±3.72 | 5.13±3.11 | 1.46±14.22 | 45.87±10.51 | 34.83±8.75 |
| IMP | 53.57% | 21.82±2.47 | 0.27±0.38 | 7.20±2.41 | 4.07±2.09 | 1.17±14.68 | 45.83±11.14 | 30.41±6.64 |
| Post-training |  |  |  |  |  |  |  |  |
| ICC |  |  | 0.13±0.54 | 7.22±4.12 | 5.22±3.16 | -0.93±17.21 | 48.23±8.78 | 34.84±7.68 |
| IMP |  |  | 0.61±0.29 | 10.13±2.93 | 2.29±1.50 | -8.62±16.85 | 39.28±7.71 | 29.83±6.00 |

SCT = score of sentence completion task, namely the average value of (positive endings—negative endings)/sum of all endings; SCT_P = the average number of positive endings in sentence completion task; SCT_N = the average number of negative endings in sentence completion task; AB = attentional bias; IAS = interpersonal anxiety scale; DASS_A = anxiety subscale in depression anxiety stress scale; IMP = interpretation modification program; ICC = interpretation control condition.

entered into a 2 × 2 repeated measurement ANOVA with time as a within-subject factor and group as a between-subject factor. The results indicated a significant time × group interaction effect, $F(1,49) = 12.84$, $p<0.01$, $\eta_p^2 = 0.21$. The main effects of time and group were also significant, $F(1,49) = 7.91$, $p<0.01$, $\eta_p^2 = 0.14$ for group, $F(1,49) = 19.27$, $p<0.01$, $\eta_p^2 = 0.28$ for time. Follow-up t-test revealed that there was no difference between IMP and ICC before training ($t(37.36) = -1.27$, $p = 0.21$, $d = -0.37$), but a significant difference appeared after the 5-day training ($t(31.98) = -3.90$, $p<0.01$, $d = -0.86$). Only IMP group showed reduced interpretation bias at post-assessment compared to pre-assessment [IMP: $t(27) = -5.15$, $p< 0.01$, $d = -1.03$; ICC: $t(22) = 0.71$, $p = 0.49$, $d = -0.14$].

To further analyze the change in participants' production of positive and negative interpretations, we did similar ANOVA with SCT_N and SCT_P as dependent variables. The results revealed a significant time × group interaction for SCT_P, $F(1,49) = 7.65$, $p <0.01$, $\eta_p^2 = 0.14$. The main effects of time and group were significant: $F(1,49) = 22.86$, $p<0.01$, $\eta_p^2 = 0.32$ for time and $F(1,49) = 4.71$, $p<0.05$, $\eta_p^2 = 0.09$ for group. Results also revealed a significant time × group interaction for SCT_N, $F(1,49) = 14.60$, $p <0.01$, $\eta_p^2 = 0.23$. The main effects of time and group were also significant: $F(1,49) = 12.02$, $p <0.01$, $\eta_p^2 = 0.20$ for time and $F(1,49) = 9.167$, $p <0.01$, $\eta_p^2 = 0.16$ for group. Independent t-tests revealed that there were no differences between ICC and IMP in production of positive and negative interpretations before training ($t(36.30) = -0.87$, $p = 0.39$, $d = -0.25$ for SCT_P, and $t(37.27) = 1.35$, $p = 0.19$, $d = 0.39$ f for SCT_N), but significant differences between the two groups showed after training: $t(49) = -2.92$, $p<0.01$, $d = 0.81$ for SCT_P, and $t(49) = 4.35$, $p<0.01$, $d = 1.18$ f for SCT_N. Paired t-tests revealed that only IMP group generated significantly more positive interpretations and less negative interpretations after training [IMP: $t(27) = -5.08$, $p<0.01$, $d = 1.09$; ICC: $t(22) = -1.59$, $p = 0.13$, $d = -0.33$ for SCT_P; IMP: $t(27) = 4.58$, $p<0.01$, $d = 1.02$; ICC: $t(22) = -0.34$, $p = 0.74$, $d = -0.07$ for SCT_N].

## Attentional bias

A 2 × 2 ANOVA revealed the marginally significant main effect of time, $F(1,47) = 4.05$, $p = 0.05$, $\eta_p^2 = 0.08$. Overall, participants had a smaller AB in the post-training test than in the pre-training test. However, the interaction effect and the main effect of group were not significant ($F(1,47) = 1.76$, $p = 0.19$, $\eta_p^2 = 0.04$ for interaction effect; $F(1,47) = 0.92$, $p = 0.34$, $\eta_p^2 = 0.02$ for the main effect of group).

## Social anxiety

For IAS, a 2 × 2 ANOVA revealed that the interaction effect of time × group was significant, $F(1,49) = 21.39$, $p<0.01$, $\eta_p^2 = 0.30$; the main effect of time was significant, $F(1,49) = 4.97$, $p<0.01$, $\eta_p^2 = 0.09$; but the main effect of group was only marginally significant, $F(1,49) = 3.41$, $p = 0.07$, $\eta_p^2 = 0.07$. Before training, there was no difference of IAS between IMP and ICC, $t(49) = 0.02$, $p = 0.10$, $d = 0.004$. As the training proceeded, a significant distinction between IMP and ICC was shown, $t(49) = 3.98$, $p<0.05$, $d = 1.11$. For paired t-tests, IMP group reported a significant decrease from pre-training assessment to post-training assessment, and increase in IAS in ICC was marginally significant [IMP: $t(27) = 4.47$, $p<0.01$, $d = 0.72$; ICC: $t(22) = -2.04$, $p = 0.05$, $d = -0.24$].

For r DASS_A, the main effect of group was significant, $F(1,49) = 5.19$, $p<0.05$, $\eta_p^2 = 0.10$. But the main effect of time and the interaction effect of time × group were not signifiicant, $F(1,49) = 0.67$, $p = 0.42$, $\eta_p^2 = 0.01$.

## Mediation effect

To explore how IMP training changed social anxiety, mediation analyses were conducted. The possible indirect effect of treatment group on social anxiety symptom reductions from baseline

to post-treatment through 1) changes in attentional bias, 2) change in SCT_P, 3) change in SCT_N, 4) change in SCT were tested. However, the 95% confidence intervals of all the mediation analyses overlapped zero, indicating no indirect effect.

## Discussion

The current study tested the efficacy of a 5-session IMP training in healthy adults with social anxiety symptoms. Results suggested that compared to control training, IMP significantly modified interpretation bias in healthy adults. By utilizing the sentence completion task, we revealed that IMP could boost the generation of positive interpretations and decrease the generation of negative interpretation in the ambiguous social scenarios and these results extended previous findings that IMP could change the endorsement of different interpretations [7]. This result was important because in the real-life situations, individuals had to generate their own interpretations instead of choosing one from several alternatives. By testing pre- and post-training social anxiety levels, we found that IMP was an effective intervention in the reduction of social anxiety in healthy adults. In early studies, researchers have illustrated with Chinese depressed adults that IMP showed more positive interpretations and fewer depression symptoms compared to ICC after training [30]; in another work, socially anxious Chinese were randomly assigned to ABM/IMP/combined training or control condition, and only IMP showed less threat interpretation and more benign interpretation than control after training. However, the change in social anxiety scores were not reported [31]. Our work extended these previous studies by showing that utilizing a Chinese version IMP can modify interpretation bias and social anxiety in healthy adults. Our work also provided primary evidence of the efficacy of interpretation bias modification using Chinese word stimuli and further studies could try to improve social anxiety in clinical or subclinical samples with these materials.

We found that the indirect effect of interpretation bias change in the relationship between training condition and improvement of social anxiety was not significant and this result did not support the hypothesis that IMP improved social anxiety by reducing interpretation bias. It might be explained by the fact that high correlation between the mediators and the predictor ($r = 0.45{\sim}0.64$, $p{<}0.01$) decreased the power of tests of mediation. Also, the current sample size was quite small for a meditation model, and as such there was likely not enough power to detect any mediation effect [32]. The lack of positive findings in our study and previous studies might be attributed to the discrepancy in the interpretation bias measurements. Studies using WSAP to measure IB generally showed a mediation effect of the change in IB [33], whereas studies using other measurements have provided both supportive and opposed evidence (positive results see [13]; negative results see [34]). The mixed results in current studies might also suggest other possible mediation effects. For example, in a preliminary investigation, researchers reported that a combination of attentional and interpretation bias training could improve rejection sensitivity and self-compassion [32], both of which were associated with social anxiety [35, 36]. Future research could also explore these mediators.

The last hypothesis of the present research was that manipulation of interpretation bias could change the attentional bias. To this end, we modified interpretation bias and investigated the change of attentional bias. Although IMP group showed decreased attentional bias, their change was not superior to the ICC group. Considering similar change in attentional bias in both groups, practice effect might be the reason for the improvement. The result that IMP did not improve attentional bias was in line with the studies conducted by Bowler et al. [17], but different from the findings presented by Amir, Bomyea and Beard [18], which revealed that attentional bias was changed even after a single session of IMP training. In Amir et al.'s study, a modified Posner procedure was conducted to measure AB. Researchers believed that the

Posner procedure could reflect the disengagement component of attention [37], but there was a general agreement that the dot-probe task was a useful measure of attentional bias as a single entity that incorporated different components of attention [38]. Thus, it was reasonable to speculate that IMP could only reduce certain component of attention processes, however, more studies were needed to explore the effect of interpretation bias training on attentional process.

This research has several limitations. The first was that we did not collect follow-up data from our participants, and the long-term effect of IMP training was not clear. Another limitation was that we chose the IAS scale to measure social anxiety. It was true that other self-rating measurements like Liebowitz Social Anxiety Scale [39] was more commonly used in this field, but the IAS showed high reliability and validity (correlation coefficient between baseline IAS and DASS_A was 0.46, $p<0.01$) in the present study and altering measurements might not influence our results. In the sentence completion task, participants were asked to generate and code sentence completions by themselves. Although participants were not aware of their assignment and IMP and ICC conditions had a high degree of similarity, it was still difficult to exclude the influence of the demand characteristics of the participants. Finally, the group size in the current study was relatively small and only health adults were included. Future studies with larger sample sizes were needed to support the results in the present study.

## Conclusion

The present study suggested that compared to control training, IMP showed more positive interpretations, less negative interpretations and reduced interpretation bias after a 5-session training. IMP also showed more reduction in social anxiety than control training. This study provided preliminary evidence for the efficacy of the Chinese version of IMP training.

## Supporting information

**S1 Checklist. CONSORT checklist.**
(DOC)

**S1 File. Data.**
(XLS)

**S2 File. Study protocol (in Chinese).**
(DOCX)

**S3 File. Study protocol (in English).**
(DOCX)

## Author Contributions

**Conceptualization:** Fan Zhang.

**Data curation:** Chenwei Huang.

**Funding acquisition:** Fan Zhang, Guanghui Deng.

**Investigation:** Chenwei Huang, Tianya Hou, Luna Sun, Yaoguang Zhou.

**Methodology:** Xiaofei Mao.

**Project administration:** Guanghui Deng.

**Supervision:** Guanghui Deng.

**Validation:** Tianya Hou.

**Writing – original draft:** Fan Zhang, Chenwei Huang.

**Writing – review & editing:** Fan Zhang, Xiaofei Mao, Tianya Hou.

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
