## [Decision Letter · Decision Letter 0]

1 Apr 2021

PONE-D-20-31856

Efficacy of the Chinese version interpretation bias modification training in an unselected sample

PLOS ONE

Dear Dr. Deng,

Thank you for submitting your manuscript to PLOS ONE. After careful consideration, we feel that it has merit but does not fully meet PLOS ONE’s publication criteria as it currently stands. Therefore, we invite you to submit a revised version of the manuscript that addresses the points raised during the review process.

The manuscript has been evaluated by three reviewers, and their comments are available below. You will see the reviewers have commented on the strengths of your manuscript. However, they have also raised a number of concerns that should be addressed before the manuscript can be further considered for publication.

The key concerns noted by the reviewers relate to the reporting of the methods. Specifically, the reviewers requested clarity regarding the randomization and allocation concealment procedures, the calculated sample size, and the methods used for multiple imputation. Reviewer 1 noted that the reporting could be improved to more closely align with the CONSORT guidelines. 

Reviewer 2 has recommended that you cite specific previously published works. As always, we recommend that you please review and evaluate the requested works to determine whether they are relevant and should be cited. It is not a requirement to cite these works. 

We look forward to receiving your revised manuscript.

Kind regards,

Danielle Poole

Staff Editor

PLOS ONE

Journal Requirements:

4. Please include a caption for each figure in the manuscript.

6. Please amend either the title on the online submission form (via Edit Submission) or the title in the manuscript so that they are identical.

7. PLOS' guidelines require  that the title should be "specific, descriptive, concise, and comprehensible to readers outside the field" (https://journals.plos.org/plosone/s/submission-guidelines#loc-title), and in this case the title may not be specific enough and not clearly understandable to readers outside the field. Please modify the title accordingly to meet PLOS standards.

Please also clarify in your submission when study recruitment and procedures took place, as the submission notes 'This study was done in the first week of October 2019' (pg 4), but the online clinical registration states that the study took place in October 2020. If your study was conducted in 2019, then please explain in your manuscript why registration to a clinical trial registry was done retrospectively (for more information about our requirements for clinical trials, please consult https://journals.plos.org/plosone/s/submission-guidelines#loc-clinical-trials).

8. Please provide corresponding author's institutional email address.

9. We suggest you thoroughly copyedit your manuscript for language usage, spelling, and grammar. If you do not know anyone who can help you do this, you may wish to consider employing a professional scientific editing service.  

Reviewers' comments:

Reviewer's Responses to Questions

**Comments to the Author**

1. Is the manuscript technically sound, and do the data support the conclusions?

Reviewer #1: Partly

Reviewer #2: Yes

Reviewer #3: Yes

2. Has the statistical analysis been performed appropriately and rigorously? 

Reviewer #1: No

Reviewer #2: Yes

Reviewer #3: Yes

3. Have the authors made all data underlying the findings in their manuscript fully available?

Reviewer #1: Yes

Reviewer #2: Yes

Reviewer #3: Yes

4. Is the manuscript presented in an intelligible fashion and written in standard English?

Reviewer #1: No

Reviewer #2: Yes

Reviewer #3: No

5. Review Comments to the Author

Reviewer #1: The objective of this prospective, randomized controlled trial (RCT) is to assess the effectiveness of the IMP training (compared to the ICC training) on evaluation of interpretation bias, social anxiety, and attentional bias, and subsequently determine possible mediation effects. The study was registered as a RCT within the Chinese Clinical Trial Registry (with a legit ChiCTR number), and was approved by the respective IRB/Ethics Committee. While the study objectives sound interesting, is important, and on target, some shortcomings were observed, in regards to abiding by the CONSORT guidelines for conducting and reporting results of high-quality randomized controlled trials (RCTs). Some other (statistical) comments were also provided.

1. Methods:

Methods reporting appeared very messy. An orderly manner is suggested, following CONSORT guidelines, without repeating information, such as Trial Design, Participant Eligibility Crtieria and settings, Interventions, Outcomes, sample size/power considerations, Interim analysis and stopping rules, Randomization (details on random number generation, allocation concealment, implementation), Blinding issues, etc, should be mentioned. The authors are advised to create separate subsections for each of the possible topics (whichever necessary), and that way produce a very clear writeup. I see the Authors indeed made an attempt; however, they are advised to write it carefully, following nice examples in the manuscript below:

https://www.sciencedirect.com/science/article/pii/S0889540619300010

Specific comments:

(a) For instance, the randomization and allocation concealment should be made very clear (they are NOT the same thing); the trial staff recruiting patients should NOT have the randomization list. Randomization should be prepared by the trial statistician, and he/she would not participate in the recruiting.

(b) The paper states: "....assigned to IMP and ICC groups with 1:1 allocation ratio." (see page 4). Was it a block randomization (BR), given that BR is often recommended in clinical studies to ensure a balance in sample size across groups over time.

https://www.ncbi.nlm.nih.gov/pmc/articles/PMC2267325/

If Yes, then what's the block size?

(c) Sample size/power: The sample size/power computations should be conducted using the primary response; it is not clear what was done. Furthermore, the sample size/power statement should include "the name of the statistical test (one- or two-sided) employed", and also the sample size number thus obtained. Those are missing.

(d) Statistical Analysis: Overall, looks straightforward. Repeated measures ANOVA was used; however, ANOVA methods are based on strict normality of the responses. Was that ascertained in the analysis?

Multiple imputation was conducted, but what method was used? Was it MICE?

https://www.ncbi.nlm.nih.gov/pmc/articles/PMC3074241/

2. Results & Conclusions:

(a) The authors should check that any statement of significance should be followed by a p-value in the entire Results section. The Results section look OK.

(b) The current study utilizes 51 patients, almost giving the feeling of a pilot study. The Conclusions section should clearly state that the results/conclusions are "only" from this Chinese population, and allude to future studies with higher sample sizes, and/or combining other populations to determine the differences.

Reviewer #2: This article reports an RCT of an interpretation training program vs. sham training on social anxiety in an unselected Chinese student sample.

Overall, my evaluation of the manuscript was positive. It represents a first step in testing out this particular interpretation training paradigm for social anxiety in a Chinese sample, as is reported appropriately as such. The reporting standards are generally good, and other strengths include the pre-registration, availability of data and protocol, and use of intention to treat analyses. I have only some relatively minor suggestions to make:

Abstract:

Say 'may be an effective way' instead of 'is an effective way' in the first sentence -efficacy is still an open question

2nd sentence needs reworking or some punctuation added to be easily understandable

Introduction:

The first paragraph needs to be a bit more cautious about the evidence for CBM, as it is a bit mixed.

Methods:

Randomization - I appreciated the detail provided by the authors, but it was still is a bit unclear. Was the allocation actually random (and if so how was it done - e.g. any blocks / stratification etc) or was it simply pseudo-random based on alternating participant numbers?

What were the predictor variables for the multiple imputation, and how many imputed datasets were created?

Results:

When describing the results for the DASS it would be useful to have the statistics and the effect sizes for the non-significant main effect of time and the timex group interaction. Given that the group x time interaction is n.s. it is not really justified to present the two t-tests comparing pre and post-training scores across the groups.

Discussion:

There are a couple of recent studies investigating different kinds of CBM for interpretation in Chinese samples, and it would be good to refer to these in the context of the relatively recent testing of CBM-I in China:

https://link.springer.com/article/10.1007/s12144-020-01094-4

https://link.springer.com/article/10.1007/s12144-020-00867-1

Reviewer #3: The authors test the efficacy of a Chinese version of interpretation bias modification training on interpretation bias, attention bias, and self-report measures of anxiety and social anxiety. Effects of interpretation bias modification training were found on interpretation bias and social anxiety, but not on attention bias and anxiety measured by the DASS. I have several comments to improve the manuscript.

1) The general aim of this manuscript was to test the efficacy of interpretation bias modification using Chinese word stimuli. However, the importance of this contribution is not well discussed in the discussion.

2) What do the authors mean by “benign interpretations”? I think positive would be more appropriate, unless I’m mistaken.

3) The authors use the word “prove” quite liberally when discussing research findings. This should be changed.

4) In addition to these wording issues, there are several other locations where the readability in English could be improved.

5) The authors state in the methods that “Then, participants coded each of their sentence completion as positive, negative or neutral…” could this effect the results? In the context of the training, it may prime certain responses or increase the likelihood of demand characteristic.

6) On page 8, the authors state “Independent t-tests revealed that there were no differences between ICC and IMP in production of positive and negative interpretations before training (ps>0.05)…” Can the authors provide the full statistics for positive and negative interpretations?

7) Similarly, on page nine when reporting the attention bias effects, the authors state “However, the interaction effect and the main effect of group were not significant (ps>0.05).” Can the full statistics for these effects (and others that may be missing) be reported?

8) The authors should briefly discuss why they think there was main effect of time on AB?

9) What was the power for the mediation tests?

6. PLOS authors have the option to publish the peer review history of their article (what does this mean?). If published, this will include your full peer review and any attached files.

Reviewer #1: No

Reviewer #2: No

Reviewer #3: No

---

## [Author Response · Author response to Decision Letter 0]

21 Apr 2021

Dear editors and reviewers,

Thank you for your helpful suggestions! After careful study of the comments, we revised the original paper carefully according to the following suggestions. The revised parts were marked in red in the related manuscript. And here are the reviewers’ comments and our reply. 

Reviewer #1: The objective of this prospective, randomized controlled trial (RCT) is to assess the effectiveness of the IMP training (compared to the ICC training) on evaluation of interpretation bias, social anxiety, and attentional bias, and subsequently determine possible mediation effects. The study was registered as a RCT within the Chinese Clinical Trial Registry (with a legit ChiCTR number), and was approved by the respective IRB/Ethics Committee. While the study objectives sound interesting, is important, and on target, some shortcomings were observed, in regards to abiding by the CONSORT guidelines for conducting and reporting results of high-quality randomized controlled trials (RCTs). Some other (statistical) comments were also provided.

1. Methods:

Methods reporting appeared very messy. An orderly manner is suggested, following CONSORT guidelines, without repeating information, such as Trial Design, Participant Eligibility Crtieria and settings, Interventions, Outcomes, sample size/power considerations, Interim analysis and stopping rules, Randomization (details on random number generation, allocation concealment, implementation), Blinding issues, etc, should be mentioned. The authors are advised to create separate subsections for each of the possible topics (whichever necessary), and that way produce a very clear writeup. I see the Authors indeed made an attempt; however, they are advised to write it carefully, following nice examples in the manuscript below:

https://www.sciencedirect.com/science/article/pii/S0889540619300010

Answer:

Thank you for the advice. We had read the recommended paper carefully and reorganized the statements in the methods part. Subtitles were also added. Please check the revised part in the methods.

Specific comments:

(a) For instance, the randomization and allocation concealment should be made very clear (they are NOT the same thing); the trial staff recruiting patients should NOT have the randomization list. Randomization should be prepared by the trial statistician, and he/she would not participate in the recruiting.

Answer:

Randomization and allocation concealment was revised as following: 

2.7 Randomization and blinding

Randomization in 1:1 allocation was determined by the following process: upon arriving at the laboratory, participants were informed of their numbers which were generated automatically by the computer. The participants and the trial staffs (SL and ZY) know the number but they were blind to the allocation. In fact, if the number was an odd number, the participant was assigned to the ICC group and if the number was an even number, the participant was assigned to the IMP group. Allocation concealment was kept until the end of the study.

(b) The paper states: "....assigned to IMP and ICC groups with 1:1 allocation ratio." (see page 4). Was it a block randomization (BR), given that BR is often recommended in clinical studies to ensure a balance in sample size across groups over time.

https://www.ncbi.nlm.nih.gov/pmc/articles/PMC2267325/

If Yes, then what's the block size?

Answer: This was not a block randomization; it was just a simple randomization without blocks. I think the original description of randomization was not clear and please see: 2.7 Randomization and blinding for the revised description.

(c) Sample size/power: The sample size/power computations should be conducted using the primary response; it is not clear what was done. Furthermore, the sample size/power statement should include "the name of the statistical test (one- or two-sided) employed", and also the sample size number thus obtained. Those are missing.

Answer: we added “2.5 Sample size calculation” to explain the sample size calculation details. Sample size was based on detecting changes in social anxiety levels, with the assumptions of a two-sided 5% significance level. To detect a large interaction effect size (0.4), the total sample size would be 52 to give a power of 80%. 

(d) Statistical Analysis: Overall, looks straightforward. Repeated measures ANOVA was used; however, ANOVA methods are based on strict normality of the responses. Was that ascertained in the analysis?

Multiple imputation was conducted, but what method was used? Was it MICE?

https://www.ncbi.nlm.nih.gov/pmc/articles/PMC3074241/

Answer:

First, the Multivariate imputation by chained equations was used to address missing values. We added the name the of the method in Data analysis.

Secondly, to test the normality of the data, Shapiro-Wilk tests were conducted and the results revealed that all the variables were consist with normal distribution except for DASS_A score. Results showed that Z=0.16, p<0.01 for pre-training DASS_A; Z=0.14, p<0.05 for post-training DASS_A. We tried several methods to normalize the data and found that reciprocal works. Z=0.11, p=0.16 for reciprocal of DASS_A before training, and Z=0.11, p=0.09 for reciprocal of DASS_A after training. Reciprocals of DASS_A were entered into ANOVA and relevant results were revised.

2. Results & Conclusions:

(a) The authors should check that any statement of significance should be followed by a p-value in the entire Results section. The Results section look OK.

Answer:

The results section was checked.

(b) The current study utilizes 51 patients, almost giving the feeling of a pilot study. The Conclusions section should clearly state that the results/conclusions are "only" from this Chinese population, and allude to future studies with higher sample sizes, and/or combining other populations to determine the differences.

Answer:

First, the limitation of small sample size was added in the last paragraph of discussion.

Secondly, the conclusion section was revised in consideration of the small sample size.

Reviewer #2: This article reports an RCT of an interpretation training program vs. sham training on social anxiety in an unselected Chinese student sample.

Overall, my evaluation of the manuscript was positive. It represents a first step in testing out this particular interpretation training paradigm for social anxiety in a Chinese sample, as is reported appropriately as such. The reporting standards are generally good, and other strengths include the pre-registration, availability of data and protocol, and use of intention to treat analyses. I have only some relatively minor suggestions to make:

Abstract:

Say 'may be an effective way' instead of 'is an effective way' in the first sentence -efficacy is still an open question

2nd sentence needs reworking or some punctuation added to be easily understandable

Answer:

Abstract of the manuscript was revised.

Introduction:

The first paragraph needs to be a bit more cautious about the evidence for CBM, as it is a bit mixed.

Answer:

The original sentence was modified as “Meta analyses of Randomized Controlled Trials (RCT) have provided early evidence for ABM and CBM-I in reducing cognitive biases and social anxiety symptoms, but the effects were moderated by characteristics of the training procedure and study design”.

Methods:

Randomization - I appreciated the detail provided by the authors, but it was still is a bit unclear. Was the allocation actually random (and if so how was it done - e.g. any blocks / stratification etc) or was it simply pseudo-random based on alternating participant numbers?

Answer:

Following this advice and the suggestion made by reviewer 1, the methods section was reorganized. And a new subtitle “2.7 Randomization and blinding” was added to describe the randomization and concealment. Randomization in 1:1 allocation was determined by the following process: upon arriving at the laboratory, participants were informed of their numbers which were generated automatically by the computer. The participants and the trial staffs (SL and ZY) know the number but they were blind to the allocation. In fact, if the number was an odd number, the participant was assigned to the ICC group and if the number was an even number, the participant was assigned to the IMP group. Allocation concealment was kept until the end of the study.

What were the predictor variables for the multiple imputation, and how many imputed datasets were created?

Answer:

Replacement values were generated for all dependent variables using multivariate imputation by chained equations based on five replications. See 2.8 data analysis for the revised description.

Results:

When describing the results for the DASS it would be useful to have the statistics and the effect sizes for the non-significant main effect of time and the timex group interaction. Given that the group x time interaction is n.s. it is not really justified to present the two t-tests comparing pre and post-training scores across the groups.

Answer:

T-tests for n.s. interaction effects were deleted.

Discussion:

There are a couple of recent studies investigating different kinds of CBM for interpretation in Chinese samples, and it would be good to refer to these in the context of the relatively recent testing of CBM-I in China:

https://link.springer.com/article/10.1007/s12144-020-01094-4

https://link.springer.com/article/10.1007/s12144-020-00867-1

https://pubmed.ncbi.nlm.nih.gov/28855880/

Answer:

Thank you for recommending the above papers. We had read them carefully and cited them. These works showed the effectiveness of CBM-I in Chinese with depression or social anxiety symptoms. Although Yang et al. (2017) found that only CBM-I group showed less threat interpretation and more benign interpretation than control after training, they did not report change in social anxiety. And the contribution of our work was that our work extended these previous studies by showing that utilizing a Chinese version IMP can modify interpretation bias and social anxiety in healthy adults. See the revised sentences in the first paragraph of discussion section.

Reviewer #3: The authors test the efficacy of a Chinese version of interpretation bias modification training on interpretation bias, attention bias, and self-report measures of anxiety and social anxiety. Effects of interpretation bias modification training were found on interpretation bias and social anxiety, but not on attention bias and anxiety measured by the DASS. I have several comments to improve the manuscript.

1) The general aim of this manuscript was to test the efficacy of interpretation bias modification using Chinese word stimuli. However, the importance of this contribution is not well discussed in the discussion.

Answer: 

This contribution was added in the last sentence of the first paragraph in discussion section.

2) What do the authors mean by “benign interpretations”? I think positive would be more appropriate, unless I’m mistaken.

Answer: 

During intervention, a benign interpretation or a negative interpretation word was showed and followed by an ambiguous sentence. “Benign” was used because some of the interpretation word was positive and some of them was neutral.

3) The authors use the word “prove” quite liberally when discussing research findings. This should be changed.

Answer: 

Thank you for the advice. All the “prove” has been replaced.

4) In addition to these wording issues, there are several other locations where the readability in English could be improved.

Answer: 

Language of the manuscript was polished carefully.

5) The authors state in the methods that “Then, participants coded each of their sentence completion as positive, negative or neutral…” could this effect the results? In the context of the training, it may prime certain responses or increase the likelihood of demand characteristic.

Answer: 

Thank you for the advice. We added the influence of demand characteristic as a limitation. 

6) On page 8, the authors state “Independent t-tests revealed that there were no differences between ICC and IMP in production of positive and negative interpretations before training (ps>0.05)…” Can the authors provide the full statistics for positive and negative interpretations?

7) Similarly, on page nine when reporting the attention bias effects, the authors state “However, the interaction effect and the main effect of group were not significant (ps>0.05).” Can the full statistics for these effects (and others that may be missing) be reported?

Answer: 

We have revised the results section and related statistics were reported.

8) The authors should briefly discuss why they think there was main effect of time on AB?

Answer: 

We think that as a variable calculated based on reaction time, the practice effect might lead to the change in AB in participants. This was discussed in the third paragraph of discussion section.

9) What was the power for the mediation tests?

Answer: For the current sample size, Power of the regression tests were 0.67, for medium effect size and two-sided 5% significance level. However, as the mediation tests did not show any significant mediation effect, this result was not added in the manuscript.

Again, thank you very much for the helpful advice. We revised the manuscript carefully following your advice and we hope that the revised manuscript could meet reviewers’ as well as the Journal’s requirement.

The authors

---

## [Decision Letter · Decision Letter 1]

14 May 2021

PONE-D-20-31856R1

Efficacy of the Chinese version interpretation bias modification training in an unselected sample: a randomized trial

PLOS ONE

Dear Dr. Deng,

Thank you for submitting your manuscript to PLOS ONE. After careful consideration, we feel that it has merit but does not fully meet PLOS ONE’s publication criteria as it currently stands. Therefore, we invite you to submit a revised version of the manuscript that addresses the points raised during the review process.

We look forward to receiving your revised manuscript.

Kind regards,

Sandra Carvalho

Academic Editor

PLOS ONE

Journal Requirements:

Reviewers' comments:

Reviewer's Responses to Questions

**Comments to the Author**

1. If the authors have adequately addressed your comments raised in a previous round of review and you feel that this manuscript is now acceptable for publication, you may indicate that here to bypass the “Comments to the Author” section, enter your conflict of interest statement in the “Confidential to Editor” section, and submit your "Accept" recommendation.

Reviewer #1: (No Response)

Reviewer #2: (No Response)

Reviewer #3: All comments have been addressed

2. Is the manuscript technically sound, and do the data support the conclusions?

Reviewer #1: Yes

Reviewer #2: Yes

Reviewer #3: (No Response)

3. Has the statistical analysis been performed appropriately and rigorously? 

Reviewer #1: Yes

Reviewer #2: Yes

Reviewer #3: (No Response)

4. Have the authors made all data underlying the findings in their manuscript fully available?

Reviewer #1: Yes

Reviewer #2: Yes

Reviewer #3: (No Response)

5. Is the manuscript presented in an intelligible fashion and written in standard English?

Reviewer #1: Yes

Reviewer #2: Yes

Reviewer #3: (No Response)

6. Review Comments to the Author

Reviewer #1: The authors addressed all my previous queries. Maybe, it will be a bit more prudent to provide a reference, or some references, to the use of reciprocals for normalization, as that is not so commonly used!

Reviewer #2: Thank you to the authors for revising their manuscript following my comments. Although the revisions were largely helpful, there were some points where something was still unclear or potentially problematic:

2.5 Sample size calculation:

This still needs a bit more information, e.g. what kind of interaction (incl. how many groups and time-points), and what kind of effect size (partial eta squared?) is the 0.4? And why was a large effect assumed?

2.6 Interim analyses and stopping guidelines

I think the authors have misunderstood this item on the CONSORT checklist- this is about the whole trial (e.g. stopping if many adverse events noticed) not individual participants.

2.7 Randomization and blinding

There is still not enough information about the randomization - how were these numbers generated by the computer? Or were the numbers just sequential, alternating odd and even, in which case the allocation is not really random (which would be quite a major limitation that would need addressing in the discussion).

"Allocation concealment was kept until the end of the study" - I think the researchers have misunderstood the term allocation concealment and should read the CONSORT elaboration document carefully.

2.8 Data analysis

It is good to have some extra information about the multiple imputation, but I am not sure I fully understand - the authors say all dependent variables were included - does this mean only post-training variables, or also baseline variables? It would be good to be more explicit about this in the paper.

Some minor points:

The sentence in the abstract "But few Chinese training procedure was available and the effect of interpretation training on attentional bias remained unclear." would read better as "However, little research had been carried out in Chinese samples, and the effect of interpretation training on other processes such as attentional bias also remained unclear.

And in the second to last sentence it would be better to say 'efficacy' rather than 'effectiveness' of the training.

In the section 2.3 Intervention

"Participants would saw a total..." should be "Participants saw a total"

In 2.8 Data analysis

" skewers distribution" should be "skewed distribution"

Reviewer #3: (No Response)

7. PLOS authors have the option to publish the peer review history of their article (what does this mean?). If published, this will include your full peer review and any attached files.

Reviewer #1: No

Reviewer #2: No

Reviewer #3: No

---

## [Author Response · Author response to Decision Letter 1]

10 Jun 2021

(respond to reviewers has been submitted as a supporting file)

Dear editors and reviewers,

Thank you for your helpful suggestions! We revised the original paper carefully according to the following suggestions. The revised parts were marked in red in the related manuscript. And here are the reviewers’ comments and our reply. 

Reviewer #1: 

The authors addressed all my previous queries. Maybe, it will be a bit more prudent to provide a reference, or some references, to the use of reciprocals for normalization, as that is not so commonly used!

Answer: 

Thank you for the advice. The methodology of Box and Cox transforms [1] was used widely to achieve normal distribution [2,3]. λ was the parameter of the transformation. This family of transformations included most of those in use by statisticians such as the square root, log transform, etc. If λ=-1, Y(λ) would equal to the reciprocals. In our study, applying the Box–Cox transformation with λ=-1 allowed obtaining a Shapiro–Wilk coefficient. Z=0.11, p=0.16 for reciprocal of DASS_A before training, and Z=0.11, p=0.09 for reciprocal of DASS_A after training. 

Reference

1. Box, G.E.P. and Cox, D.R. (1964). An Analysis of Transformations. Journal of the Royal Statistical Society: Series B (Methodological), 26: 211-243. doi:10.1111/j.2517-6161.1964.tb00553.x

2. Casal C.A., Anguera M.T., Maneiro R., Losada J.L. (2019). Possession in Football: More Than a Quantitative Aspect – A Mixed Method Study. Front Psychol,18;10:501. doi:10.3389/fpsyg.2019.00501.

3. Gecili E.，Ziady A.，Szczesniak R.D. (2021). Forecasting COVID-19 confirmed cases, deaths and recoveries: Revisiting established time series modeling through novel applications for the USA and Italy. PLoS One. 2021,16(1):e0244173. doi: 10.1371/journal.pone.0244173.

Reviewer #2:

Thank you to the authors for revising their manuscript following my comments. Although the revisions were largely helpful, there were some points where something was still unclear or potentially problematic:

2.5 Sample size calculation:

This still needs a bit more information, e.g. what kind of interaction (incl. how many groups and time-points), and what kind of effect size (partial eta squared?) is the 0.4? And why was a large effect assumed?

Answer: 

Thank you for the advice. Sample calculation was modified as:

G*Power software version 3.1.9.2 was used to estimate required sample size for this study. This study used repeated measurement analysis of variance (ANOVA) to analyze time, group and interaction effects of the interventions. Therefore F-test (ANOVA repeated measurement, within-between interactions) was chosen. Sample size was based on detecting changes in social anxiety levels. In line with the previous study[1], effect size (ηp2) was set at 0.23 and alpha value was set at 0.05. Approximately 40 participants (i.e., 20 participants in each arm) would provide 80% power and 52 participants (i.e., 26 participants in each arm) would provide 90% power to detect a statistical significance.

Here, the power of 80%[2] or 90%[3] has both been widely used and we provide the results of sample calculation under both conditions.

References:

1 Amir, N., & Taylor, C. (2012). Interpretation Training in Individuals with Generalized Social Anxiety Disorder: A Randomized Controlled Trial. J consult clin psycho, 80, 497-511. doi: 10.1037/a0026928

2 Antognelli SL, Sharrock MJ, Newby JM. (2020). A randomised controlled trial of computerised interpretation bias modification for health anxiety. J Behav Ther Exp Psychiatry, 66:101518. doi: 10.1016/j.jbtep.2019.101518.

3 Moore B, Dudley D, Woodcock S. (2019). The effects of martial arts participation on mental and psychosocial health outcomes: a randomised controlled trial of a secondary school-based mental health promotion program. BMC Psychol, 7(1):60. doi: 10.1186/s40359-019-0329-5.

I think the authors have misunderstood this item on the CONSORT checklist- this is about the whole trial (e.g. stopping if many adverse events noticed) not individual participants.

Answer: 

Thank you for the advice. We relearned the CONSORT checklist and found that the “interim analyses and stopping guidelines” was not applicable for the present study. We are very grateful to the reviewer for reminding us of use of stopping guidelines. 

2.7 Randomization and blinding

There is still not enough information about the randomization - how were these numbers generated by the computer? Or were the numbers just sequential, alternating odd and even, in which case the allocation is not really random (which would be quite a major limitation that would need addressing in the discussion).

"Allocation concealment was kept until the end of the study" - I think the researchers have misunderstood the term allocation concealment and should read the CONSORT elaboration document carefully.

Answer: 

The random number (ranged from 1 to 100) was generated by the Microsoft Excel using the random number function. This detailed information was added in the manuscript.

According to CONSORT, allocation concealment seeks to prevent selection bias. But in the current study, post-training tests were self-rating scales which would be influenced by demand characteristics of the participants. As a result, participants and the experimenter did not know group assignment during intervention assignment and pre-/post-training tests. In line with the CONSORT checklist as well as the reality, we revised the sentence to “Allocation concealment was kept until interventions and post-training tests were finished.”

2.8 Data analysis

It is good to have some extra information about the multiple imputation, but I am not sure I fully understand - the authors say all dependent variables were included - does this mean only post-training variables, or also baseline variables? It would be good to be more explicit about this in the paper.

Answer: 

As shown in Figure 1, one participant forgot to take post-training tests of SCT, AB and IAS and his missing data were generated by multiple imputation. The rest of the data were complete. See 2.8 Data analysis for the modified parts.

Some minor points:

The sentence in the abstract "But few Chinese training procedure was available and the effect of interpretation training on attentional bias remained unclear." would read better as "However, little research had been carried out in Chinese samples, and the effect of interpretation training on other processes such as attentional bias also remained unclear.

And in the second to last sentence it would be better to say 'efficacy' rather than 'effectiveness' of the training.

In the section 2.3 Intervention

"Participants would saw a total..." should be "Participants saw a total"

In 2.8 Data analysis

" skewers distribution" should be "skewed distribution"

Answer: 

Thank you for the advice. The sentences and words were revised and marked in the manuscription.

Again, thank you very much for the helpful advice. We revised the manuscript carefully following your advice and we hope that the revised manuscript could meet reviewers’ as well as the Journal’s requirement.

---

## [Editor Report · Decision Letter 2]

13 Jul 2021

Efficacy of the Chinese version interpretation bias modification training in an unselected sample: a randomized trial

PONE-D-20-31856R2

Dear Dr. Deng,

We’re pleased to inform you that your manuscript has been judged scientifically suitable for publication and will be formally accepted for publication once it meets all outstanding technical requirements.

Kind regards,

Sandra Carvalho

Academic Editor

PLOS ONE
---

## [Editor Report · Acceptance letter]

19 Jul 2021

PONE-D-20-31856R2 

Efficacy of the Chinese version interpretation bias modification training in an unselected sample: a randomized trial 

Dear Dr. Deng:

I'm pleased to inform you that your manuscript has been deemed suitable for publication in PLOS ONE. Congratulations! Your manuscript is now with our production department. 

Kind regards, 

on behalf of

Dr. Sandra Carvalho 

Academic Editor

PLOS ONE